# Biosynthesis of the highly oxygenated tetracyclic core skeleton of Taxol

Chengshuai Yang [1,6], Yan Wang[1,6], Zhen Su[1,2,6], Lunyi Xiong[1,2], Pingping Wang[1], Wen Lei[3], Xing Yan [1] ✉, Dawei Ma [4] ✉, Guoping Zhao [1,5] ✉ & Zhihua Zhou [1] ✉

Taxol is a widely-applied anticancer drug that inhibits microtubule dynamics in actively replicating cells. Although a minimum 19-step biosynthetic pathway has been proposed and 16 enzymes likely involved have been characterized, stepwise biosynthetic reactions from the well-characterized di-oxygenated taxoids to Taxol tetracyclic core skeleton are yet to be elucidated. Here, we uncover the biosynthetic pathways for a few tri-oxygenated taxoids via confirming the critical reaction order of the second and third hydroxylation steps, unearth a taxoid 9$\alpha$-hydroxylase catalyzing the fourth hydroxylation, and identify CYP725A55 catalyzing the oxetane ester formation via a cascade oxidation-concerted acyl rearrangement mechanism. After identifying a acetyltransferase catalyzing the formation of C7-OAc, the pathway producing the highly-oxygenated 1$\beta$-dehydroxybaccatin VI with the Taxol tetracyclic core skeleton is elucidated and its complete biosynthesis from taxa-4(20),11(12)-diene-5$\alpha$-ol is achieved in an engineered yeast. These systematic studies lay the foundation for the complete elucidation of the biosynthetic pathway of Taxol.

Taxol (paclitaxel), belonging to the diterpenoid alkaloid, was first obtained from the bark of *Taxus brevifolia*[1] and, so far, has been one of the most potent and widely applied anticancer drugs[2]. During the past two decades, enormous efforts including chemical total synthesis[3–7], biosynthesis using cell suspension cultures of *Taxus* species[8], and semisynthetic route using baccatin III or 10-deacetyl baccatin III isolated from needles of *Taxus* species trees as starting material[9], have been made to relieve the short supply of Taxol due to its low accumulation in *Taxus* plants (~0.01% dry weight)[10]. Meanwhile, due to the complexity of biogenesis for its special core skeleton expansion and various modifications including oxidation and acylation, elucidation of Taxol biosynthetic pathway and revealing its key enzymatic reaction mechanisms have been an attractive challenge for both academia and industry.

The biosynthesis of Taxol is presumed to originate from the cyclization of geranylgeranyl diphosphate (GGPP) (the common precursor of diterpenoids). By incubating tritium-labeled GGPP with a *Taxus* bark extract, radiochemical taxa-4(5),11(12)-diene (**1**) was produced[11]. The incubation of tritium-labeled **1** with *Taxus* stem sections resulted in the production of radiochemical 10-deacetylbaccatin III and taxol, demonstrating that **1** should be the key diterpenoid core skeleton of Taxol[11]. Besides, the feeding tritium-labeled taxa-4(20),11(12)-diene-5$\alpha$-ol (**2**) has also been testified to be converted to 10-deacetylbaccatin III and Taxol in *Taxus* plant tissue[12]. Based on these above studies, **2** has been popularly proposed as the mono-oxygenated intermediate of Taxol[13–15].

Over the years, numerous efforts were also made in employing enzymatic (or chemical) catalysis to synthesize the probable

[1]Key Laboratories of Plant Design and Synthetic Biology, CAS Center for Excellence in Molecular Plant Sciences, Institute of Plant Physiology and Ecology, Chinese Academy of Sciences, Shanghai, China. [2]University of Chinese Academy of Sciences, Beijing, China. [3]Shanghai Research Institute of Chemical Industry, Shanghai, China. [4]State Key Laboratory of Chemical Biology, Shanghai Institute of Organic Chemistry, Chinese Academy of Sciences, Shanghai, China. [5]Institute of Synthetic Biology, Shenzhen Institutes of Advanced Technology, Chinese Academy of Sciences, Shenzhen, China. [6]These authors contributed equally: Chengshuai Yang, Yan Wang, Zhen Su. ✉e-mail: yanxing@cemps.ac.cn; madw@mail.sioc.ac.cn; gpzhao@sibs.ac.cn; zhouzhihua@cemps.ac.cn

intermediates (or related functional groups) which were naturally identified or proposed to be related to the Taxol biosynthetic pathway. To this date, at least 19 catalyzing steps including minimum nine oxidation reactions mainly mediated by cytochrome P450s (P450s) and five acylation modification steps catalyzed by acyltransferases were proposed to involve in this pathway[16]. Of which, the P450s for the hydroxylation of C5, C10, C13, C7, and C2[17–21], the acyltransferases for the acylation of C5-OH, C10-OH, C2-OH, and C13-OH[22–27], as well as the enzymes for the C2′ hydroxylation and C3′-NH2 acylation of the β-phenylalanoyl-modified side chain[28–30], have been cloned and characterized over the years (Fig. 1, Supplementary Table 1, and Supplementary Data 1).

The first breakthrough in rebuilding the de novo biosynthetic pathway of Taxol was to synthesize 1 from GGPP catalyzed by taxa-4(5),11(12)-diene synthase (TS) in a recombinant *E. coli* strain[14]. Employing the same strategy of metabolic engineering, the mono-oxygenated taxoid 2 was synthesized from 1 catalyzed by taxoid 5α-hydroxylase (T5OH)[14]. Later, 5α-acetoxytaxa-4(20),11(12)-diene-10β-ol (4), a di-oxygenated taxoid, was produced in cell factories with addition of C5 acetylation catalyzed by a taxoid 5-O-acetyltransferase (T5AT) followed by C10 oxidation catalyzed by a taxoid 10β-hydroxylase (T10OH)[31]. However, no any downstream intermediate from 4 has ever been reported since then. The expected combinatorial modification using T10OH and taxoid 13α-hydroxylase (T13OH) to produce the hypothetical intermediates, in particular the tri-

oxygenated taxoids with oxidation at C5, C10, and C13, was unsuccessful[16,19]. On the other hand, although the formation of di-oxygenated taxoid taxa-4(20),11(12)-diene-5α,13α-diol (5) from intermediate 2 catalyzed by T13OH was reported as early as in 2001[19], no further oxidized modification of 5 has ever been published. Therefore, biosynthetic pathways even for the tri-oxygenated taxoids need to be further clarified or elucidated.

The biosynthesis of the tetra-oxygenated taxoids is proposed to be initiated via hydroxylation modification at C9 catalyzed by taxoid 9α-hydroxylase. Although a putative cDNA clone for taxoid 9α-hydroxylase was claimed[15], neither its sequence nor further characterization has ever been reported. Besides, hydroxylation modifications at C2, C7, and C1 were assumed to happen in the later steps for the synthesis of the highly oxygenated intermediates of Taxol[15]. The characterized taxoid 2α-hydroxylase (T2OH) and 7β-hydroxylase (T7OH) were known to hydroxylate a tetra-oxygenated taxoid taxusin to produce 2α- and 7β-hydroxytaxusin, respectively, and further, reciprocally, catalyze the hydroxylation of the corresponding 7β- and 2α-hydroxytaxusin to form the intermediate hexa-oxygenated 2α,7β-dihydroxytaxusin (13) (Fig. 1)[20,21]. However, taxusin and its derivatives (7β-hydroxytaxusin, 2α-hydroxytaxusin and 2α,7β-dihydroxytaxusin), have never been regarded as intermediates in the previously proposed biosynthetic pathway of Taxol[20,21]. Thus, the enzymes responsible for C1 and C9 hydroxylation modifications, and the reaction order of hydroxylation modifications at C2, C7, and C1 in the biosynthesis of

**Fig. 1 | Summary of previously reported and supposed reactions involved in the biosynthesis of taxoids.** GGPP geranylgeranyl diphosphate, TS taxa-4(5),11(12)-diene synthase, T5OH taxoid 5α-hydroxylase, TCPR *Taxus* cytochrome P450 reductase, T5AT and TAX19 taxoid 5-O-acetyltransferase, T13OH taxoid 13α-hydroxylase, T10OH taxoid 10β-hydroxylase, T9OH taxoid 9α-hydroxylase, T2OH taxoid 2α-hydroxylase, T7OH taxoid 7β-hydroxylase, T1OH taxoid 1β-hydroxylase, TBT taxoid 2α-O-benzoyltransferase, DBAT 10-deacetylbaccatin 10β-O-

acetyltransferase, BAPT 13-O-(3-amino-3-phenylpropanoyl) transferase, T2′OH taxoid 2′-hydroxylase, DBTNBT N-debenzoyl-2′-deoxytaxol N-benzoyltransferase, PAM phenylalanine aminomutase. Dashed lines represent the proposed steps and unknown enzymes were marked in red. New formed groups in compounds were marked with red. Tri-oxygenated taxoids were speculated as the hypothetical intermediates and hydroxylation modification at C9 catalyzed by T9OH was proposed as the fourth oxidation reaction step in the biosynthetic pathway of Taxol.

Taxol were unclear. In addition, the key step for Taxol biosynthesis, i.e., the formation of Taxol tetracyclic core skeleton with an oxetane ester is unknown although this core skeleton was proposed to form following the formation of an acetylated taxadiene-2,5,7,9,10,13-hexaol intermediate before the formation of baccatin III[32], which was employed as the precursor for the industrial semi-synthesis of Taxol[9].

In this work, we re-characterize the enzymes, the previously reported P450s, i.e., T10OH[18] and T13OH[19], as well as acetyltransferases T5AT[22] and TAX19[33], via substrate feeding of *Saccharomyces cerevisiae* cells expressing corresponding genes (in vivo reaction) and substrate incubation with crude protein lysate from the corresponding *S. cerevisiae* cells (in vitro reaction). With further identification of several intermediates including di-oxygenated taxoids 5α-acetoxytaxa-4(20),11(12)-diene-13α-ol (**6**) and diacetyl taxoid 5α,13α-diacetoxytaxa-4(20),11(12)-diene (**9**), the first three hydroxylation steps are uncovered, leading to the formation of tri-oxygenated taxoid 5α,13α-diacetoxytaxa-4(20),11(12)-diene-10β-ol (**10**). Furthermore, from a *Taxus* cDNA library, we discover two CYP450 genes, of which, *CYP725A37* encoded taxoid 9α-hydroxylase catalyzes the fourth hydroxylation step, forming a tetra-oxygenated taxoid (9,10-deacetyltaxusin, **11**). The CYP725A55 catalyzes 2α-benzoyloxy-7β-acetoxytaxusin (**15**) to form the Taxol tetracyclic core skeleton of 1β-dehydroxybaccatin VI (**16**) via a cascade oxidation-concerted acyl rearrangement process. Based on these results, the complete biosynthetic pathway of 1β-dehydroxybaccatin VI (**16**), the highly oxygenated Taxol intermediate sharing the same Taxol tetracyclic core skeleton as that of baccatin III and Taxol, is elucidated and its biosynthesis from **2** is further confirmed in engineered yeast.

## Results

### Biosynthesis of tri-oxygenated taxoids

The hydroxylation modifications at C10 and C13 of the mono-oxygenated taxoid taxa-4(20),11(12)-diene-5α-ol (**2**) were predicted as the second or third oxidation steps in the biosynthetic pathway of Taxol[15]. However, the hydroxylation modification of the di-oxygenated taxoid 5α-acetoxytaxa-4(20),11(12)-diene-10β-ol (**4**) seems to come to a dead end and the research progress on hydroxylation and acylation modifications of the alternative di-oxygenated taxoid taxa-4(20),11(12)-diene-5α,13α-diol (**5**) was limited. Specifically, there are six potential combined reaction orders (A-F) to synthesize tri-oxygenated taxoids from **2** catalyzed by T5AT, T10OH and T13OH (Supplementary Table 2). Previous studies demonstrated that the efforts to produce hypothetical tri-oxygenated taxoids via these potential reaction orders were either unsuccessful for A, C, D, and F[16,19] or hard for B[16,19], probably due to the conversion rate of T13OH towards 5α-acetoxytaxa-4(20),11(12)-diene (**3**) was too low to identify the structure of the resulted product. Therefore, we preferentially tried the untested route E.

Co-expression of *T13OH* and *TCPR* encoding a cytochrome P450 reductase from *Taxus* in *S. cerevisiae* resulted in the formation of a metabolite in both in vivo and in vitro reactions using **2** as the substrate (Supplementary Fig. 1a–c, e). This metabolite was concluded as **5** with a hydroxyl group at C13 based on spectra analyses including high-resolution electrospray ionization mass spectrometry (HR-ESIMS) and nuclear magnetic resonance (NMR) spectroscopy including $^1$H, $^{13}$C, etc (Supplementary Fig. 1d, f, g). These above results confirmed that T13OH could efficiently install a hydroxyl group at C13 position of the mono-oxygenated taxoid **2** to produce the di-oxygenated taxoid **5**. Next, T5AT catalyzed **5** to produce a metabolite assigned as an acetylated derivative 5α-acetoxytaxa-4(20),11(12)-diene-13α-ol (**6**) (Supplementary Fig. 2), indicating that T5AT transferred an acetyl group towards the C5-OH of **5**. Later, the co-expression of *T10OH* and *TCPR* in *S. cerevisiae* and feeding the di-oxygenated taxoid **6** resulted in a compound which was confirmed to be a tri-oxygenated taxoid 5α-acetoxytaxa-4(20),11(12)-diene-10β,13α-diol (**7**) (Supplementary Fig. 3),

suggesting T10OH could install a hydroxyl group at C10 position of **6**. These results confirmed that the reaction order E is a practicable biosynthetic route for converting **2** to **7** through C13 hydroxylation, C5-OH acetylation, and C10 hydroxylation in three sequentially steps. The previous unsuccessful combination F was demonstrated to be accessible when using another acetyltransferase TAX19 instead of T5AT but maybe regarded as minor pathway in the biosynthetic of **7** from **5** because of the low conversion rate of TAX19 towards **8** (Supplementary Fig. 4).

Besides, taking consideration of the fact that TAX19 prefers to add acetyl group towards the east-west pole positions at C5-OH and C13-OH of taxoids[33], we employed **6** as the substrate for both in vivo and in vitro reactions using *S. cerevisiae* strain expressing *TAX19*. A metabolite was produced and further assigned as a diacetyl taxoid, 5α,13α-diacetoxytaxa-4(20),11(12)-diene (**9**) (Supplementary Fig. 5). This result revealed TAX19 could add an acetyl group towards the C13-OH of **6**. When the di-oxygenated taxoid **9** was employed as the substrate for both in vivo and in vitro reactions with a *S. cerevisiae* strain expressing *T10OH* and *TCPR*, the consumption of **9** and formation of another metabolite were observed (Supplementary Fig. 6a, b). This metabolite was further assigned as the tri-oxygenated taxoid 5α,13α-diacetoxytaxa-4(20),11(12)-diene-10β-ol (**10**), suggesting T10OH introduces a hydroxyl group at C10 position of **9** (Supplementary Fig. 6c–e). Another taxoid 10β-hydroxylase (T10OH2) exhibited the similar function as T10OH (Supplementary Fig. 7).

We realized the biosynthesis of a few tri-oxygenated taxoids from **2** via re-characterization of the known enzymes and confirming the second and third oxygenated reaction order (Fig. 2a). Besides, to the best of our knowledge, compounds **6**, **7**, and **10** in biosynthetic routes of tri-oxygenated taxoids have never been reported previously (Supplementary Table 3).

### Taxoid 9α hydroxylase for tetra-oxygenated taxoids

The tetra-oxygenated taxoids 9,10-deacetyl taxusin (**11**) and taxusin (**12**) have been isolated from *Taxus* plants previously[34]. However, their biosynthetic pathways are still unknown. Compared with the structure of **10**, the C9 position was oxidized in both **11** and **12**, which was proposed to be catalyzed by a taxoid 9α-hydroxylase belonging to CYP725A subfamily[16]. In order to characterize the functions of CYP725As, we tried to screen the targeted taxoid 9α-hydroxylase through in vivo reactions using the *S. cerevisiae* strains harboring a series of uncharacterized *CYP725As*. A cDNA database based on *Taxus* EST datasets collected from NCBI and RNA-seq datasets generated in this work was established. 26 uncharacterized *CYP725As* from the cDNAs of *Taxus* twigs were respectively co-expressed with *TCPR* in *S. cerevisiae*. The co-expression of *CYP725A37* and *TCPR* in *S. cerevisiae* resulted in the formation of a metabolite in both in vivo and in vitro reactions using **10** as the substrate (Fig. 2b and Supplementary Fig. 8a). According to HR-ESIMS and NMR spectroscopy, the structure of this metabolite was elucidated as a tetra-oxygenated taxoid, 9,10-deacetyl taxusin (**11**), which is the C9 hydroxylation product of **10** (Fig. 2c, d and Supplementary Fig. 8b). These results confirmed CYP725A37 was a taxoid 9α-hydroxylase to install a hydroxyl group at C9 positon of taxoids. Based on the identified taxoid 9α-hydroxylase CYP725A37, the biosynthetic pathway from **2** to **11** could be elucidated involving five sequentially steps including C13 hydroxylation, C5 acetylation, C13 acetylation, C10 hydroxylation and C9 hydroxylation (Fig. 2).

Compared with the structure of **11**, **12** has the same oxidation level but its C9-OH and C10-OH were both acetylated. Thus, the biosynthesis of **12** could be realized via the further acylation of intermediate **11**. T5AT has been demonstrated to preferentially acetylate the "northern" hemisphere hydroxyls at C9 and C10 towards tetra-oxygenated taxoids previously[33]. When **11** was employed as the substrate for both in vivo and in vitro reactions with the *S. cerevisiae* strain expressing *T5AT*, HPLC analysis demonstrated that **12** was formed, accompanied with

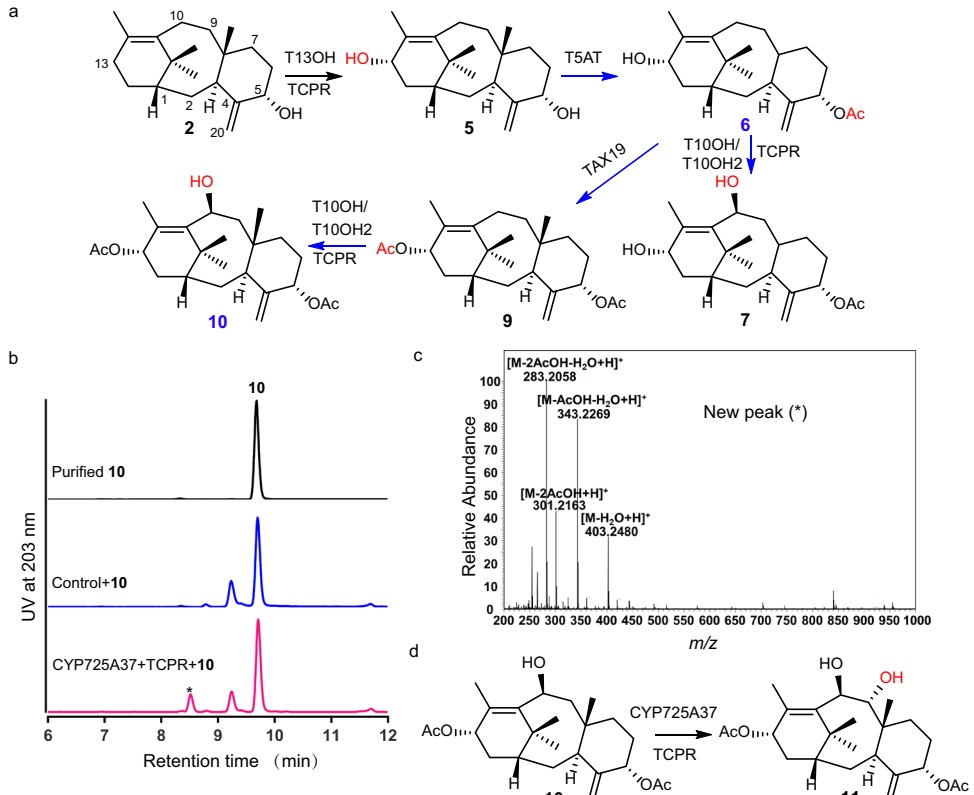

**Fig. 2 | The biosynthetic pathways of tri-oxygenated and tetra-oxygenated taxoids and characterization of the taxoid 9α hydroxylase. a** The biosynthetic pathways of tri-oxygenated taxoids. Reaction steps and new compounds characterized in this study were marked in blue. New formed groups in compounds were marked with red. **b** HPLC analysis of in vivo reaction for *S. cerevisiae* strain YCYP725A37 expressing CYP725A37 and TCPR using 5α,13α-diacetoxytaxa-4(20),11(12)-diene-10β-ol (**10**) as the substrate. Substrate was marked with compound number and product was marked with an asterisk. *S. cerevisiae* strain YTCPR expressing TCPR was used as the control. **c** Mass spectra (ESI) of new peak (*). These experiments were performed three times with similar results each time. **d** The C9 hydroxylation reaction catalyzed by CYP725A37 to produce 9,10-deacetyl taxusin (**11**) from **10**. Source data are provided as a Source Data file.

proposed monoacetylated **11** at C9-OH or C10-OH (Supplementary Fig. 8c–g), confirming T5AT could add acetyl groups towards the C9-OH and C10-OH of **11** to form **12**. Besides, we testified that T7OH and T2OH could convert **12** to yield a hexa-oxygenated taxoid 2α,7β-dihydroxytaxusin (**13**), which has been reported in a previous study[20]. Thus, we could elucidate a complete biosynthetic pathway of **13** from **2** which involves eight sequentially steps based on the confirmed biosynthetic pathway of tri-oxygenated taxoids, unearthed taxoid 9α-hydroxylase and re-characterization of the reported enzymes.

### Formation of Taxol tetracyclic core skeleton

The hexa-oxygenated intermediate **13** lacks the Taxol tetracyclic core skeleton with an oxetane ester in baccatin III and Taxol as well as their derivatives. P450 was suggested to participate in the formation of oxetane ester in the Taxol tetracyclic core skeleton, especially for the formation of a proposed 4β, 20-epoxide intermediate in a plausible mechanism for the oxetane ester formation[32]. However, no enzyme has been reported to catalyze the relative reaction until now. According to the structures of taxoids with the oxetane ester group isolated from *Taxus* plants, we speculated that the preferential substrate for the oxetane ester formation might be 2α-benzoyloxy-7β-acetoxytaxusin (**15**) and then semi-synthesized **15** from **13** (Supplementary Fig. 9a, b, g), which was used as the substrate to screen the left uncharacterized *CYP725As* through in vivo reaction. The co-expression of *CYP725A5S* and *TCPR* in *S. cerevisiae* resulted in the formation of two metabolites (Peaks A and B) (Fig. 3a). The structure of one metabolite (peak A) was deduced as 1β-dehydroxybaccatin VI (**16**) based on its 1D NMR data, being consistent with the related NMR

spectrum of **16** reported in literature[35] (Fig. 3a, b, e and Supplementary Fig. 10e). The other metabolite (peak B) with the same molecular formula as **16**, was assigned as a C4-C20 epoxide derivative 2-deacetyl-2α-benzoylbaccatin I (**17**) based on the HR-ESIMS and NMR spectroscopy (Fig. 3a, c, e and Supplementary Fig. 10f). These results demonstrated that CYP725A55 was responsible for the oxetane ester formation to generate the highly oxygenated Taxol tetracyclic core skeleton in Taxol biosynthesis.

### The mechanism of oxetane ester formation

In 1987, Potier and coworkers proposed that the oxetane and tertiary C4 ester of the plant-isolated taxoids was formed via enzyme-catalyzed epoxidation of C4-C20 double bond and subsequently acyl group rearrangement by analyzing the structure of isolated taxoids bearing the oxetane ester motif and its possible allyl alcohol precursors[36] (Supplementary Fig. 9c). Decades later, Walker and coworkers reported the 10-deacetylbaccatin 10β-O-acetyltransferase (DBAT) could catalyze the acylation of the tertiary C4-OH[26]. Based on the above results, the formation of the oxetane ring and the acetylation of C4-OH catalyzed by P450 and acetyltransferase were proposed as another stepwise pathway to the biosynthesis of oxetane ester. However, none of the aforementioned pathways has been verified because no enzyme-catalyzed oxetane ester formation has been reported to date.

In this study, deuterium labeled compound 5α-trideuterated acetyl-2α-benzoyloxy-7β-acetoxy taxusin (**20**) was synthesized from **5** via the combination of chemosynthetic and biosynthetic strategies and employed as the substrate in an in vivo reaction using *S. cerevisiae* strain co-expressing *CYP725A5S* and *TCPR* (Supplementary Fig. 9d–g).

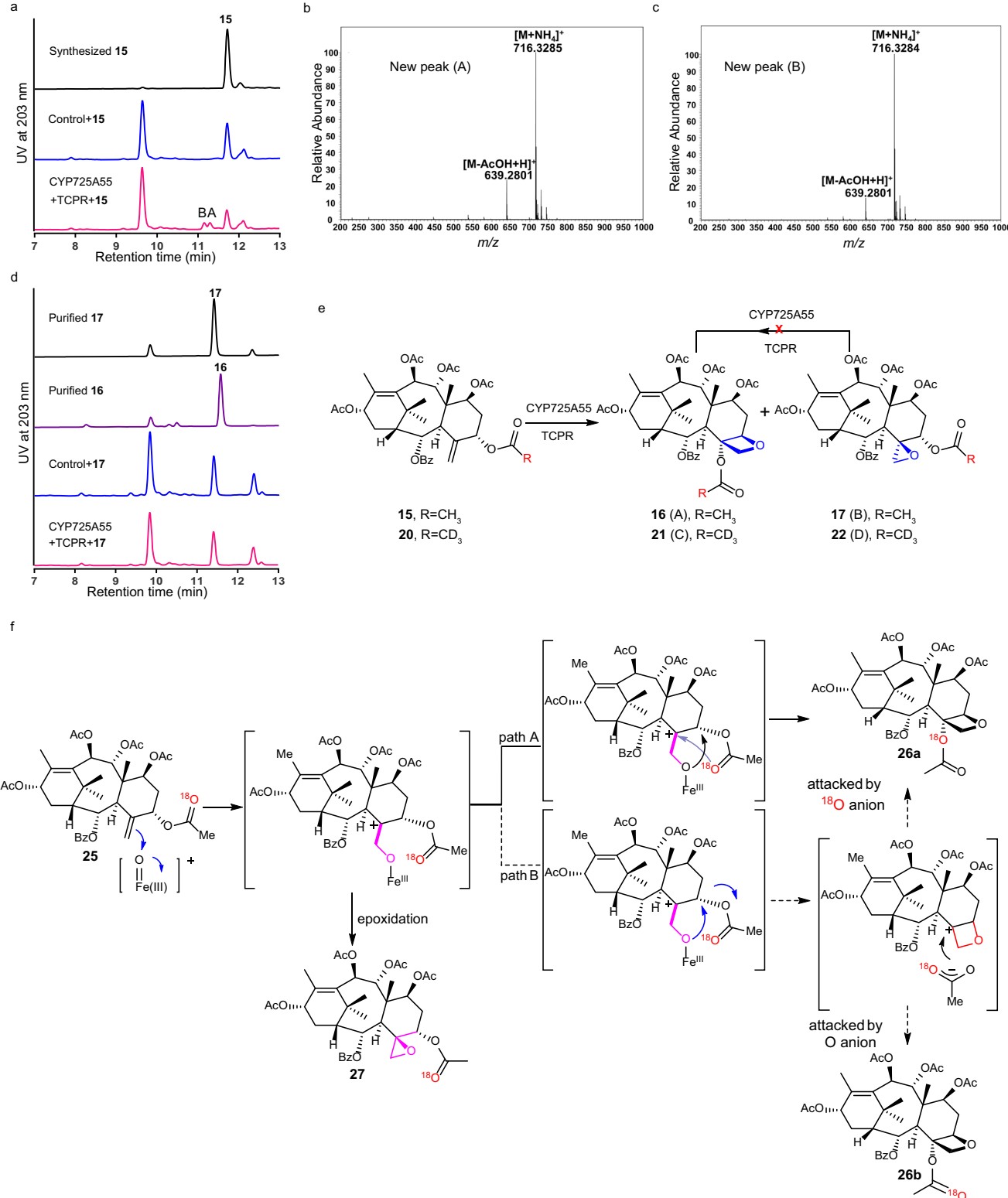

**Fig. 3 | The formation of Taxol tetracyclic core skeleton with an oxetane ester.**
**a** HPLC analysis of in vivo reaction for *S. cerevisiae* strain YCYP725A55 expressing CYP725A55 and TCPR using 2α-benzoyloxy-7β-acetoxytaxusin (**15**) as the substrate. Substrate was marked with compound number and products were marked with A and B. *S. cerevisiae* strain YTCPR expressing TCPR was used as the control. **b** Mass spectra (ESI) of new peak (A). **c** Mass spectra (ESI) of new peak (B). **d** HPLC analysis of in vivo reaction for *S. cerevisiae* strain YCYP725A55 expressing CYP725A55 and TCPR using 2-deacetyl-2α-benzoylbaccatin I (**17**) as the substrate. Substrate was

marked with compound number. *S. cerevisiae* strain YTCPR expressing TCPR was used as the control. These experiments were performed three times with similar results each time. **e** Oxidation reaction of hexa-oxygenated taxoid derivatives catalyzed by CYP725A55. Labeled or regular methyl was marked with red. Arrow with a cross indicated this reaction was failed. **f** Proposed process of the formation of oxetane ester. [18]O was marked with red. Source data are provided as a Source Data file.

Two compounds were detected (Peaks C and D in Supplementary Fig. 10a). Compounds 1β-dehydroxy-4α-trideuterated acetyl-baccatin VI (**21**, peak C) with oxetane ester and 5α-trideuterated acetyl-2-deacetyl-2-benzoylbaccatin I (**22**, peak D) with the epoxy ring were isolated and characterized by HR-ESIMS and NMR spectra analyses (Supplementary Fig. 10b–f). A significant $^2$H signal detected at δ 2.26 ppm for **21** indicates that there was only one deuterated acetyl group. Comparing the $^1$H NMR spectra of oxetane ester compound **16** with the deuterated **21**, the signal of methyl group in C4-acetoxyl disappeared which indicated that the C4-acetoxyl group in the oxetane product was derived from the C5-acetoxyl group of 5α-trideuterated derivative **20** (Supplementary Fig. 10d, e). To further verify whether epoxy ester was involved in the generation of oxtane ester via rearrangement, the epoxy derivatives (**17** and **22**) were employed in the in vivo reaction using *S. cerevisiae* strain co-expressing *CYP725A55* and *TCPR*. However, no oxetane ester product was detected by HPLC analysis (Fig. 3d and Supplementary Fig. 10g, h), implying that the epoxide ester may not be the precursor for the biosynthesis of oxetane ester. Based on these experimental results, the oxetane ester is likely to be formed via a direct oxidation-acyl rearrangement process without the formation of epoxide ester. To further identify whether the acyl rearrangement occurred through concerted or stepwise process or both of them. The C5-acetyl $^{18}$O-labeled 2α-benzoyloxy-7β-acetoxytaxusin (**25**) was prepared (Supplementary Fig. 11a, b), and employed as substrate in in vivo reaction using the *S. cerevisiae* strain expressing *CYP725A55* and *TCPR*. The $^{18}$O-labeled oxetane derivatives **26** (**26a** or the mixture of **26a** and **26b**) could be formed (Supplementary Fig. 11c), which further confirmed the C4 acetoxyl group of oxetane ester **26** was derived from the C5 acetoxyl group of **25**. All of the $^{18}$O in $^{18}$O-labelled oxetane derivatives **26** was reserved in the deacetyl product $^{18}$O-labeled 4,7,9,10,13-pentadeacetyl-1β-dehydroxybaccatinVI (**28a**) based on the MS analysis, indicating that the acyl rearrangement overwhelmingly occurred through a concerted process (Fig. 3f and Supplementary Fig. 11d–g).

### Biosynthesis of 1β-dehydroxybaccatin VI

In order to elucidate the complete biosynthetic pathway to 1β-dehydroxybaccatin VI (**16**) sharing the same Taxol tetracyclic core skeleton as that of baccatin III, Taxol and their derivatives, we need to further find an enzyme responsible for the acylation of C7-OH of taxoid **13** to produce **15**. We cloned and expressed 9 uncharacterized acyltransferases encoding genes from our cDNA database in *E.coli*, and obtained *E.coli* strains YAT1-9. Then we screened these acyltransferases by incubating **13**, benzoyl-CoA, acetyl-CoA, and the combined crude enzymes from the *S. cerevisiae* strain YTBT expressing 2α-O-benzoyl transferase (TBT)[24] and each of the *E.coli* strains YAT1-9 in in vitro reaction, respectively. The strainYAT5 expressing *AT5* resulted in the production of **15**, demonstrating acyltransferase AT5 could acetylate the C7-OH of taxoid **13** (Supplementary Fig. 12a–c). Based on the identified P450s in this work and the acyltransferase AT5 as well as the re-characterization of several previously reported enzymes[16,18,19,22,24,33], biosynthetic pathway of **16** from **2** was elucidated.

Next, we constructed a *S. cerevisiae* strain named as YDBVI to biosynthesize **16** via co-expressing 12 genes from *Taxus* plant, including *T13OH*, *T5AT*, *TAX19*, *T10OH*, *CYP725A37*, *T2OH*, *T7OH*, *AAE4* (encoding enzyme AAE4 catalyzing benzoic acid to produce benzoyl-CoA)[37], *TBT*, *AT5*, *CYP725A55*, and *TCPR* (Fig. 4a). Our target compound **16** could be detected by liquid chromatography-mass spectrometry (LC–MS) after feeding **2** to the strain YDBVI (Fig. 4b). Interestingly, although compound **16** is the predominant product in the culture of YDBVI using **2** as substrate (Fig. 4b), **16** appeared in a 1:1 ratio with **17** in the feeding studies using **15** as the substrate (Fig. 3). We speculate that these seemingly contradictory results might due to the complexity of the in vivo experimental system, i.e., other pathways for **16** from **2** in the YDBVI strain or the complex redox environment in the engineered strains expressing 12 genes.

Almost all of the supposed intermediates in this elucidated biosynthetic pathway were detected in the sequential co-expressing experiment (Supplementary Fig. 12d). And only the compound **11** detected from *Taxus* plants previously[34] was missed in these intermediates produced by these engineered yeasts. This result confirmed our elucidated biosynthetic pathway of **16** in this study (Fig. 4a) and illustrated that we have successfully achieved the biosynthesis of **16** from **2**.

## Discussion

Over the past two decades, numerous efforts employing chemistry, phytochemistry, biochemistry and molecular biology have been undertaken to reveal the biosynthetic pathway for Taxol. In particular, an overwhelming majority of the genes (12 in 14) involved in the biosynthetic pathway of Taxol were identified by Prof. Rodney Croteau group[15]. However, except the biosynthesis of the key diterpenoid core skeleton, taxadiene, and the following two steps of oxygenation, little progress leading to the tri-oxygenated taxoid intermediates and beyond has been made until this study. In this study, we focused on the downstream steps of **2**, which was the proposed first hydroxylation product of taxa-4(5),11(12)-diene. Followed by step-wise re-characterizing key enzymes, identifying uncharacterized enzymes via integrated assays of in vivo substrate feeding of cells and in vitro substrate incubation with crude protein lysates based on their corresponding gene-expressed *S. cerevisiae* cells, pathways for biosynthesis of the highly oxygenated tetracyclic core skeleton of Taxol, 1β-dehydroxybaccatin VI (**16**) from **2** was elucidated (Fig. 4). Along with the reviewing process of this manuscript, similar or alternative catalysis processes for the oxetane ring formation were published by three independent studies[38–40], of which, the study of Jiang et al[40]. not only supported our findings of enzyme characterization but also proved that this biosynthesis pathway network elucidated via *S. cerevisiae* chassis cells is largely similar to that via the *tobacco* chassis.

While realizing that the commonly focused di-oxygenated taxoid **4** might not be the key intermediate for the biosynthesis of Taxol, we retreated back to hypothesizing the di-oxygenated taxoid **5** transformed from mono-oxygenated taxoid **2** as the key intermediate. We re-characterized the function of several previously reported P450s and acetyltransferases[16,18,19,22,33], revealed their reaction orders via both in vitro and in vivo catalysis (Fig. 4a and Supplementary Fig. 1a, b), and finally the tri-oxygenated taxoid derivatives **7, 8**, and **10** biosynthesized from **5** were identified (Fig. 2a).

We further identified taxoid C9α-hydroxylase CYP725A37 catalyzing the formation of tetra-oxygenated taxoid **11** from **10** (Fig. 2d). Then, via re-characterizing the function of T5AT, we revealed a biosynthetic pathway for the formation of tetra-oxygenated intermediate **12** (Fig. 4a), an abundant metabolite in the heartwood of yew but long been ignored in Taxol biosynthesis[20,21]. Thus, the previously reported formation of the hexa-oxygenated taxoid intermediate **13** from **12** was confirmed in the Taxol biosynthesis pathway[20] (Fig. 4a). Furthermore, we proved that the hexa-oxygenated taxoid **13** can be acetylated by a missing acetyltransferase AT5 and the previously reported TBT[24] to form **15** (Fig. 4a).

The oxetane motif in Taxol is essential for the binding with α,β-tubulin dimer in the process of inhibiting microtubule dynamics in actively replicating cells[41]. So far, in this proposed and rebuilt synthetic pathway from **2** to **15**, all of the taxoid intermediates share the same diterpenoid taxadiene core skeleton with increasing levels of oxygenation and various acylation modifications. We identified CYP725A55 which transforms **15** to 1β-dehydroxybaccatin VI (**16**) via the critical enzymatic oxetane ester formation reaction (Fig. 3). This highly oxygenated Taxol intermediate **16** shares the same Taxol tetracyclic core skeleton as that of the well-characterized baccatin III and all Taxol/Taxol derivatives (Fig. 4a).

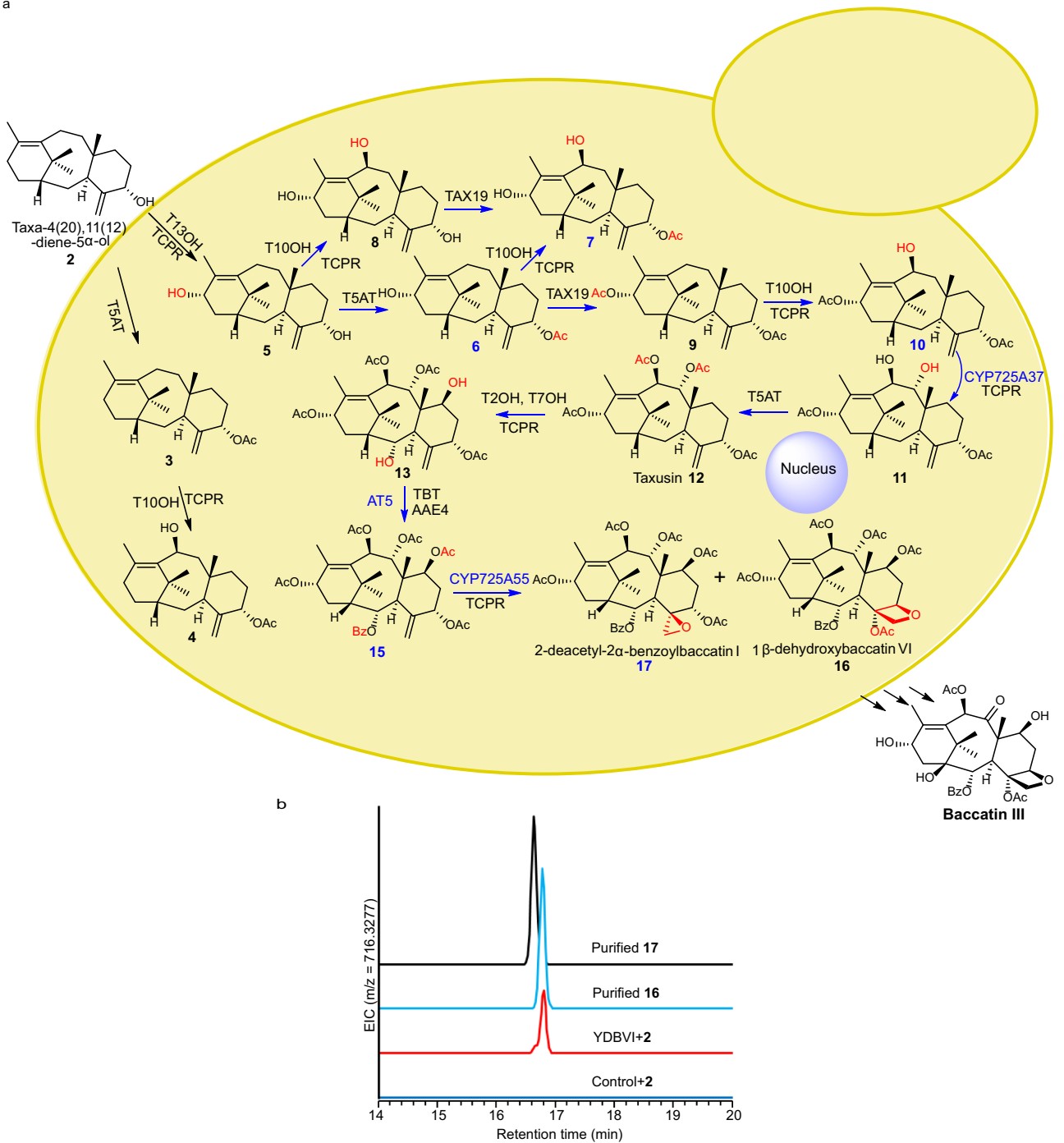

**Fig. 4 | The elucidated pathway of 1β-dehydroxybaccatin VI (16) and its biosynthesis in engineered yeast. a** The biosynthetic schematic of **16** from **2** in engineered yeasts. Reactions, enzymes and new compounds characterized in this study were marked in blue. New formed groups in compounds were marked with red. **b** LC–MS chromatograms show the formation of **16** in engineered yeast YDBVI co-expressing *T13OH*, *T5AT*, *TAX19*, *T10OH*, *CYP725A37*, *T2OH*, *T7OH*, *AAE4*, *TBT*, *AT5*, *CYP725A55*, and *TCPR*. AAE4, benzoyl-CoA synthetase. *S. cerevisiae* chassis strain YBD80 was used as the control. EIC extracted ion chromatogram. Source data are provided as a Source Data file.

Although either epoxidation-acyl rearrangement or stepwise oxetane formation–acylation process was proposed over the years for the biosynthesis of oxetane ester[26,36], neither has ever been experimentally proved. The isotope labeling ($^2$H or $^{18}$O) and feeding experiments suggest a different catalyzing process in our enzymatic reaction system (Fig. 3f). The C4-C20 double bond was oxidized by CYP725A55 to form a C4 cation intermediate, which was transferred to oxetane ester 1β-dehydroxybaccatin VI (**16**) via acyl rearrangement or captured by C20 oxygen to generate the epoxide derivative 2-deacetyl-2α-benzoylbaccatin I (**17**) (Fig. 3e, f). For the second acyl rearrangement step

to form oxetane ring, there are two possible mechanisms, either concerted or stepwise processes. The oxygen isotope experiment was carried out to characterize this process. Thematically, the $^{18}$O-labeled acetyl rearranges to the C4 hydroxyl group to form only one oxetane product **26a** via a concerted process, while both of the labeled and regular oxygen atom can equally attack the C4 cation to form two different oxetane derivatives (**26a** and **26b**) with the $^{18}$O at the different positions via a stepwise process. According to the MS analysis of the de-acetyl product **28a**, one of the oxygens was labeled and almost no regular oxygen atom attacked product was formed, which is consist

with the concerted rearrangement mechanism (Fig. 3f and Supplementary Fig. 11). These results indicated that the oxetane ester was overwhelmingly formed via a cascade oxidation-concerted acyl rearrangement mechanism in our enzymatic reaction system (Fig. 3f). It is interesting to notice that Jiang et al. also proposed an oxetane ring formation mechanism of oxidation-acyl rearrangement without using epoxide as an intermediate based on density functional theory (DFT) calculations[40], supporting our cascade oxidation-concerted acyl rearrangement mechanism.

The rebuilding of biosynthetic pathway of Taxol has already achieved the de novo biosynthesis of **2** in *E. coli* strains[14]. In this synthetic pathway from **2** to **16**, intermediates, **6, 10** and **15** have not been reported in plants until now, while **11, 12**, and **16** were isolated from *Taxus* plants previously (Supplementary Table 3)[34]. The targeted compound **16** and almost all of the supposed intermediates including **6, 9, 10, 12, 13**, and **15** could been detected by sequentially co-expressing these 12 involved enzymes, testifying the elucidated biosynthetic pathway of **16** from **2** in living cells (Supplementary Fig. 12d). The intermediate **11** was exceptionally undetected in the metabolites of these recombinant yeast strains, which may result from its efficient acetylation catalyzed by T5AT as observed in the in vivo reaction (Supplementary Fig. 8c).

It is particularly interesting to notice that, in contrast to baccatin III, which bears a hydroxy, a ketone and a hydroxy at C7, C9 and C13 positions, respectively, compound **16** bears an acetyl group at each of these three positions. It seems that the formation of some intermediates need be protected by acetylation at C7, C9 and C13 positions and then the deacetylation at C7, C9, and C13 of **16** would be involved in the downstream biosynthesis of baccatin III from **16**. The adding-and removing-protective-group strategy has been well applied in chemical synthesis and also found in the biosynthetic pathways of some complex plant natural products to protect unstable intermediates and inhibit the formation of byproducts via acetylation and deacetylation. For example, in the biosynthetic pathway of noscapine, an acetyl group was introduced at C8 by an O-acetyltransferase AT1 to function as a protective group to avoid hemiacetal formation in the next oxidation step by CYP82X1 and subsequently hydrolyzed by a carboxylesterases CXE1[42]. And an acetyl group of the intermediate vinorine serves as a stabilizing group to prevent ring-opening and was then hydrolyzed by an acetylajmalan esterase at the last stage of ajmaline biosynthetic pathway[43]. Similarly, an acetyl group of C21 introduced by acetyltransferase CsL21AT possibly acted as protecting group to stabilize the hemiacetal ring, which led to an increased yield of target kihadalactone A[44]. Moreover, acetylation might also play important roles in the transport or storage of intermediates in the biosynthetic pathway[42]. So, these introduced acetyl groups at C7, C9, and C13 may also stabilize these highly oxygenated intermediates to avoid unexpected rearrangement reactions in the biosynthesis of Taxol and related taxoids or benefit the transport of these intermediates.

Several studies have found that much higher yield of 5(12)-oxa-3(11)-cyclotaxane (OCT) or iso-oxa-cyclotaxane (iso-OCT) (85% to 95% in total) rather than **2** (5% to 15%) were produced by heterogeneously expressing T5OH in different chassis, such as *S. cerevisiae, E. coli* or *Nicotiana benthamiana*[45,46]. Structurally, different from the eight carbon-B ring system in taxoids, OCT and iso-OCT have two fused rings with C3(11) bond. It seems highly improbable to disrupt the C3(11) bond and transfer 3(11)-cyclotaxane core to the taxa-11(12)-ene core[47]. Although it seems these OCTs may only be the bio-precursors of C3(11)-cyclotaxanes[34,47], there is still a possibility that the downstream products of OCTs can rearrange to taxoids in the complex biological environments in *Taxus* plants. It is worthy to study the downstream products of OCTs and the relationship of 3(11)-cyclotaxane core and taxa-11(12)-ene core. Recently, T5OH was proposed to catalyze oxetane ring formation via the rearrangement of unstable epoxide intermediate from endotaxadiene and exotaxadiene based on the detection of a taxologenic oxetane among the found four products catalyzed by T5OH[38]. In another study[39], taxologenic oxetane formation was also proposed to be catalyzed by T5OH together with a 2-oxoglutarate/Fe(II)-dependent dioxygenase. Although these proposed mechanisms, different from that of the cascade oxidation-concerted acyl rearrangement catalyzed by CYP725A55 shown in this study and Jiang et al.[40], need further experimental confirmations, the presence of both multi-products of T5OH and different mechanisms potential for oxetane ring formation implicated that the biosynthetic pathway of Taxol is more likely a complex, matrix rather than a linear pathway. Future comprehensive verifications are needed.

In summary, our systematic studies clearly elucidated the complete biosynthetic route to the highly oxygenated 1β-dehydroxybaccatin VI with the Taxol tetracyclic core skeleton from the mono-oxygenated taxoid taxa-4(20),11(12)-diene-5α-ol, laying down a solid foundation for the breakthrough of complete elucidation of the biosynthetic pathway of Taxol. Of course, further efforts are required to elucidate the complete biosynthetic pathway of baccatin III from 1β-dehydroxybaccatin VI and other parallel biosynthetic pathway of Taxol. It is also interesting to further confirm whether Taxoid 1β-hydroxylase and 9α-dioxygenase are necessary[39] or unwanted[40] to biosynthesize baccatin III in *N. benthamiana* and *Taxus* in the future. While an important experience learnt from the recent emergence of breakthroughs[38-40], including this study, cross-disciplinary investigations and collaborations, particularly between the chemists and biochemists with the phytochemists and plant molecular physiologists are essential in order to completely fulfill this long-awaiting endeavor.

## Methods
### Media, strains, and chemicals
*Saccharomyces cerevisiae* strain BY4742 was purchased from EURO-SCARF. These engineered strains were cultured in SC minimal medium containing 6.7 g/L yeast nitrogen base without amino acids (291940, BD), 20 g/L glucose (10010518, Sinopharm Chemical Reagent) with corresponding deficient amino acid or YPD containing 10 g/L yeast extract (LP0021B, Thermo Fisher Scientific), 20 g/L bacteriological peptone (LP0137, Thermo Fisher Scientific), and 20 g/L glucose at 30°C. *Escherichia coli* TOP10 and BL21 (DE3) were cultured in Luria-Bertani (LB) medium containing 5 g/L yeast extract, 10 g/L peptone (LP0042B, Thermo Fisher) and 10 g/L sodium chloride (10019318, Sinopharm Chemical Reagent) at 37 °C. Agar (10000582, Sinopharm Chemical Reagent) was added for SC minimal medium, YPD and LB plates with 20 g/L. Primers used in this research were ordered from Sangon Biotech (Shanghai) and listed in Supplementary Data 2. Ethyl acetate (G23272C), petroleum ether (G84208B), tetrahydrofuran (011342848), triethylchlorosilane (92440B), 4-dimethylaminopyridine (14766 A), chromatography silica gel (G72651A), triethyl amine (17471H), triethylchlorosilane (92440B), pyridine (G14230A), tetra-butylammonium fluoride (14363 G), CDCl$_3$(87070 S), CD$_3$OD (84556 S), D$_6$-DMSO (38114 L), Pd(OAc)$_2$ (BD01121063), 1,4-Benzoquinone (12960B), and anisole (011072489) were purchased from Shanghai Titan Scientific. Dichloromethane (80047318), MeOH(10014118), EtOH (10009218), acetic anhydride (10000318), ethylene diamine tetraacetic acid (10009717), acetic acid (10000218), *n*-hexane (80068618), sodium bicarbonate (01-S0181), sodium sulfate (10020518), hydrazine hydrate (C196715000), HCl (10011018), and KOH (10017018) were purchased from Sinopharm Chemical Reagent. Dimethyl sulfoxide (A600163), ampicillin (A100339), dithiothreitol (A620058), and benzylsulfonyl fluoride (A610425) were purchased from Sangon Biotech (Shanghai). D$_6$-acetic anhydride (175641) and acetyl-CoA (A2056) were purchased from Sigma-Aldrich. 1,3-Dicyclo-hexylcarbodiimide (N806920) was purchased from Macklin. C$_6$D$_6$ (B100912) was purchased from Aladdin. Benzoyl CoA (ZL-23055) was purchased from Shanghai ZZBIO. Nicotinamide adenine dinucleotide

phosphate (60302ES02) was purchased from Yeasen Biotechnology (Shanghai). The high performance liquid chromatography (HPLC) grade acetonitrile (CAEQ-4-003306-4000) was purchased from Anpel Laboratory Technologies (Shanghai).

## Taxus pan-transcriptome analysis

A total of 43 publicly available transcriptome/RNA-seq datasets from *Taxus* species (https://www.ncbi.nlm.nih.gov/Taxonomy/Browser/wwwtax.cgi?id=25628) were collected from National Center for Biotechnology Information SRA (Supplementary Data 3). Nine initial assemblies were performed for the same sequencing platform of the datasets within same species. For Illumina sequencing platform data assembly, these raw sequencing datasets were processed using Trimmomatic[48] (version 0.38) to remove adapters and trim low-quality sequences with the following parameters (ILLUMINACLIP: NexteraPE_TruSeq3-PE-2.fa:2:30:10 SLIDINGWINDOW:4:15 LEADING:3 TRAILING:3 MINLEN:36). Subsequently, assembly was performed using Trinity[49] (version 2.9.0) with the following parameters (--max_memory 300 G --min_contig_length 200 --CPU 40 --bflyCPU 40 --inchworm_cpu 40) to obtain initial assembly results. Finally, seven Trinity assembly datasets were obtained. For 454 life science sequencing platform data assembly, the NCBI SRA data files were converted to SFF (Standard Flowgram Format) using sratoolkit[50] (version 2.10.5). Then the SFF files were assembled using Newbler (version 3.0, Roche) with "-m -cpu 64 -cdna" parameter to generate assembled isotigs Transcripts. To merge these assemblies, the seven Trinity assembly datasets and two Newbler assembly datasets were combined. The contigs was shredded into fragments of 2000 bp length with 200 bp overlap. The resulted fragments assembly was performed using Newbler (version 3.0, Roche) with the following parameters: -o Taxus -m -cpu 64 -cdna. For deduplication, the initial nine assembly datasets and the merged assembly datasets were combined. CD-HIT-EST[51] (version 4.8.1, parameters: -c 1 -aS 1 -M 0 -G 0 -d 0 -r 1 -g 1 -T 40 -i Transcripts.fasta -o cdhit) was utilized to obtain the final non-redundant *Taxus* transcriptome. For transcript completeness assessment, sequence similarity searches were conducted using the Diamond[52] (version 0.9.36) alignment against the Uniprot Swiss-Prot database[53] (Database version: 202001). Trinity[49] (version 2.9.0) analyze_blastPlus_topHit_coverage tool was employed to assess the completeness of non-redundant transcripts. Finally, coding sequences of the transcripts were predicted using TransDecoder(version 5.0.5) with default parameters.

## RNA-seq and de novo transcriptome assembly

*Taxus yunnanensis* Cheng et L. K. Fu twigs (including wood and needles) were homogenized using a chilled mortar and pestle, and total RNA was isolated using RNAprep Pure Plant Kit (DP441, TIANGEN BIOTECH) according to the manufacturer's instructions. For PacBio ISO-SEQ data analysis, SMRTlink (version 8.0.0, Pacific Biosciences) was used to filter and process raw subreads, using the Iso-seq3 pipeline to obtain highly accurate long reads. These raw subreads were used to generate circular consensus sequence (CCS) reads using the following parameters: --min-rq 0.9. The resulting CCS reads were then stripped of adapter sequences using LIMA. The poly-A tails were then removed using the Refine tool. Full-length non-chimeric (FLNC) reads were classified based on the presence or absence of 5' primer, 3' primer and the poly(A) tail. FLNC reads were clustered using the Iso-seq3 cluster module in the SMRTlink to generate consensus sequences, which were subsequently polished using the raw subreads. For Illumina RNA-seq data analysis, these raw reads were extracted from the sequencing results, and FastQC (Version 0.11.9) with default parameters was used to evaluate quality of the raw reads. Adaptor sequences in the raw reads were trimmed using Trimmomatic[48] (Version 0.38). Moreover, the bases at the begin and end of each reads with quality below 3 were cut. The

Trimmomatic slides from the 5' end in windows and the window size was 4 bases. When the average quality in the window is lower than the setting threshold of 15, the read is cut. Only the reads with length >36 bp was kept. The obtained data was subjected for de novo transcriptome assembly using Trinity[49] (Version 2.9.0).

## Cloning of candidate genes

RNA isolated from *Taxus yunnanensis* Cheng et L. K. Fu twigs was converted into cDNA using the PrimeScript RT reagent Kit with gDNA eraser (RR047A, Takara). In general, the open reading frames (ORFs) of candidate genes were cloned from *Taxus* twig cDNA with PrimeSTAR HS DNA Polymerase kit (R010A, Takara). Then PCR products were linked into the pMD18T vector (D101A, Takara) and transformed into *E. coli* TOP10 cells. Transformants were plated for selection on LB plates with 100 μg/mL ampicillin and positive colonies were verified by colony PCR and Sanger sequencing.

## Construction of yeast strains

*S. cerevisiae* codon-optimized genes for GGPP synthase (GGPPs), TS, T13OH, T10OH, T7OH, T5AT, TAX19, CYP725A37, TCPR, AAE4, and TBT were synthesized by Tsingke Biotechnology or Shanghai Generay Biotech and were listed in Supplementary Data 4. *S. cerevisiae* and *E. coli* strains used in this study were listed in Supplementary Table 4. PrimeSTAR HS DNA Polymerase kit was used for all PCR amplification steps according to the manufacturer's instruction. HiPure Gel DNA Pure Mini Kit (D2111-03, Guangzhou Magen Biotechnology) was used to purify DNA products according to the manufacturer's instruction. The details for construction of yeast strains can be found in Supplementary Note 1. While the overall strategy is as following:

(1) The strong promoters (GAL1p and GAL10p) and terminators commonly used in yeast metabolic engineering were chosen to express heterologous gene. In order to change the galactose-induced promoters (GAL1p and GAL10p) into constitutive expression, the GAL-promoter negative regulator, *GAL80* of BY4742 was deleted with an URA3 marker and the resulted strain YBD80 was chosen as the chassis cell for heterologous expression of candidate genes.

(2) Three selective markers, *i.e.*, HIS3, LEU and KanMX, were used for the construction of genetic engineered yeast strains based on the chassis strain YBD80 and the details were described in the Supplementary Table 4. Because multiple genes could be simultaneously integrated in the chromosome of yeast, we achieved the integration of 12 genes by three-round integrations in chassis strain YBD80 using 3 different makers (HIS3, LEU and KanMX), without recycling usage of the markers.

(3) For clone verification, in case of integration of one or two genes, five colonies were picked for verification, while in the case of integration of multiple genes (>3), 20 colonies were picked for verification.

## In vivo feeding assays

A single colony of yeast strain was incubated in 1 mL YPD medium and then grown at 30 °C, 250 rpm. Substrate (25 or 500 μM in dimethyl sulfoxide, exact concentrations for each individual assays were listed in the Supplementary Table 5) was added to the medium at 24 h. Metabolite was extracted using 1 mL ethyl acetate (EA) at 48 h and the pooled extracts were evaporated. The residue was dissolved in EA and analyzed by Gas chromatography flame ionization detector (GC-FID) or HPLC. The control strain (YBD80 or YTCPR) was treated in parallel. The concentration of product was estimated on the conversion rate and the assumption that the substrate and corresponding product shared the same chromatographic response (the estimated product concentrations for each of the reactions were shown in the Supplementary Table 5).

## In vitro enzymatic assays

In order to eliminate the effect of time-consuming ultracentrifugation and redissolution in the process of microsomes preparation on the activity of enzyme, crude protein lysate of yeast was used for these in vitro enzymatic assays. A single colony of yeast strain was incubated in 50 mL YPD medium and grown 48 h at 30 °C, 250 rpm. Then yeast cell was incubated in 200 mL YPD medium after centrifugation at room temperature and grown 24 h at 30 °C, 250 rpm. Next, yeast cell was incubated in 200 mL YPD medium after centrifugation at 4 °C and grown 24 h at 16 °C, 110 rpm. Finally, collected yeast cell was pulverized with liquid nitrogen and resuspended with 100 mM potassium phosphate buffer (pH 7.4) containing 1 mM ethylene diamine tetraacetic acid, 0.5 mM dithiothreitol, 1 mM benzylsulfonyl fluoride. After removing cell fragments using centrifugation, the supernatant was used as crude protein lysate. These in vitro enzymatic assays were conducted in 300 μl reactions of containing 100 mM potassium phosphate buffer (pH 7.4), 1 mM NADPH (for P450s) or acetyl-CoA (for acyltransferases), 25 or 500 μM substrate, and 12 mg crude protein lysate. The crude protein lysate from control strain (YBD80 or YTCPR) was treated in parallel. For the preparation of crude protein of acyltransferases, a single colony of *E. coli* strain was incubated in 4 mL LB medium and grown 12 h at 37 °C, 200 rpm. Then *E. coli* strain was incubated in 50 mL LB medium and grown at 37 °C, 200 rpm until OD600 was 0.6. Next, protein expression was induced by 0.2 mM isopropyl $\beta$-D-1-thiogalactopyranoside at 16 °C, 110 rpm for 18 h. The collected cells were suspended in 100 mM phosphate buffer (pH 7.4) and then disrupted with an Ultrasonic Cell Disruption System (JY92-IIDN, Shanghai Jingxin Industrial Development) (300 W, ultrasonic time: 3 s; rest time: 3 s, 10 min). After removing the cell debris by centrifugation and the supernatant was used for enzymatic assays. These in vitro enzymatic assays of acyltransferases were conducted in 600 μL reactions of containing 100 mM potassium phosphate buffer (pH 7.4), 1 mM acetyl-CoA, 1 mM benzoyl CoA, 500 μM substrate, 12 mg crude protein lysate from YTBT and 1 mg *E. coli* crude protein lysate. The crude protein lysate from control *E. coli* strain with empty vector was treated in parallel. The reaction system was mixed at 30 °C for 12 h. The reactants were extracted with equal volume EA and analyzed by GC-FID or HPLC.

## GC-FID analysis

GC-FID analysis was performed on a GC2010 pro (Shimadzu, Suzhou, China) equipped with a flame ionization detector using a TG-5MS capillary column (30 m × 0.25 mm × 0.25 μm, Thermo Fisher Scientific). The oven temperature was kept at 150 °C for 2 min, increased to 250 °C at a rate of 10 °C/min, held 250 °C for 20 min, then increased to 280 °C at a rate of 10 °C/min, and finally held 280 °C for 5 min. The temperature of inlet and detection were 250 °C and 260 °C, respectively. The sample volume of per injection was 2 μL in split ratio of 20. The flow rate of carrier gas (nitrogen) was kept at 1.25 mL/min.

## HPLC analysis

HPLC analysis was performed on a Shimadzu LC 20 A system (Shimadzu, Kyoto, Japan) equipped with a diode array detector, an autosampler, a binary pump and a thermostatically controlled column compartment using a Welch Boltimate™ C18 column (2.7 μm, 2.1 mm × 100 mm). Water (A) and acetonitrile (B) were used in the gradient elution system. The gradient elution program was carried out as follows: 0-2 min (15% B), 15 min (98% B), and 15.01-17 min (15% B). The flow rate was kept at 0.45 mL/min and the temperature of column compartment was set as 35 °C. Products were detected at 203 nm.

## HPLC/ESIMS analysis

The sample preparation for HPLC/ESIMS analysis is same to GC-FID or HPLC analysis. The chromatographic analysis was carried out on a Dionex Ultimate 3000 RSLC (HPG) ultra-performance liquid chromatography system (Thermo Fisher Scientific) using a Welch Boltimate™ C18 column (2.7 μm, 2.1 mm × 100 mm). Water (A) and acetonitrile (B) were used in the gradient elution system. The injection volume was 1 μL. The gradient elution program was carried out as follows: 0−1 min (5% B), 26 min (95% B), and 26.01−28 min (95% B), 28.01−30 min (5% B). The flow rate was kept at 0.35 mL/min and the temperature of column compartment was set as 35 °C. Products were also detected at 203 nm. The HR-ESIMS data were acquired using a Q Exactive quadrupole orbitrap high-resolution mass spectrometry with a HESI ionization source (spray voltage 3.5 kV, Thermo Fisher Scientific) in the positive ionization mode. The sheath and aux gas were maintained at 40 and 10 (arbitrary units), respectively. The capillary temperature and heater temperature were set at 300 °C and 350 °C, respectively. The S-Lens RF level was set at 50. The Orbitrap mass analyzer was operated at a resolving power of 70,000 in full-scan mode (scan range: 200−1000 *m/z*; automatic gain control (AGC) target: 1e6) with a dynamic exclusion setting of 4.0 s. Xcalibur 4.2 software was used for compound analysis. The number of technical and/or biological replicates was indicated in the Figure legend. For the MS analysis of biosynthetic intermediates in yeast cell, the MS analysis of purified standards were performed at the same time. Characteristic ion fragment and retention time were used to identify compounds.

## Chemical synthesis of 2

Taxa-4(5), 11(12)-diene (**1**) was extracted from the medium of strain YTA with petroleum ether (PE) and purified with chromatography silica gel using PE as eluent. The solution of **1** (2.0 g, 7.4 mmol), Pd(OAc)₂ (412 mg, 25 mol%), 1,4-Benzoquinone (1.76 g, 14.4 mmol), anisole (3.2 mL, 29.6 mmol) in acetic acid (200 mL) was degassing and charged with N₂, then the system was stirred at 50 °C overnight[54]. The reaction system was cooled to room temperature to quench the reaction. Water (300 mL) and *n*-hexane (500 mL) were added and the organic phase was separated, and the aqueous phase was extracted with *n*-hexane (3 × 500 mL). The combined organic phase was washed with saturated sodium bicarbonate, brine, dried with anhydrous sodium sulfate and filtered. The solvent was removed in vacuum and the resulted residue was purified with chromatography silica gel (PE/EA = 50/1) to obtain 5α-acetoxyl-taxa-4(20),11(12)-diene (**3**) as yellow oil (800 mg).

To the solution of 5α-acetoxyl-taxa-4(20),11(12)-diene (800 mg, 2.42 mmol) in THF /MeOH (1/1, 10 mL), KOH (1.0 g, 1.8 M in final reaction system) was added in one portion, the resulted reaction system was stirred at room temperature for 24 h. The solvent was removed in vacuum and the resulted residue was purified with chromatography silica gel (PE/EA = 30/1) to obtain taxa-4(20),11(12)-diene-5α-ol (**2**) as colorless solid (588 mg, 33% yield for 2 steps).

## Chemical synthesis of 5α-trideuterated acetoxyl taxa-4(20),11(12)-diene-13β-ol (18)

Triethyl amine (0.69 mL, 10.2 mmol), triethylchlorosilane (0.43 mL, 2.3 mmol) was dropped to the solution of taxa-4(20),11(12)-diene-5α,13β-diol (**5**) (650 mg, 2.1 mmol) in dichloromethane (15 mL) at 0 °C, subsequently. Then 4-dimethylaminopyridine (131 mg, 1.0 mmol) was added in one portion and the resulted reaction system was stirred at room temperature with 2 h. Then water was added to quench the reaction and separated. The aqueous phase was extracted with dichloromethane (3 × 30 mL), the combined organic phase was washed with bine, dried with sodium sulfate, filtered and concentrated in vacuum. The resulted residue was purified on chromatography silica gel (PE/EA = 20/1) to get the compound 13α-triethylsiloxytaxa-4(20),11(12)-diene-5α-ol (**29**) (300 mg, 34% yield).

The mixture of compound **29** (300 mg, 0.72 mmol), pyridine (2 mL) and D₆-acetic anhydride (2 mL) was stirred at room temperature overnight and the concentrated in vacuum, the residue was purified on chromatography silica gel (PE/EA = 20/1) to provide the compound 5α-trideuterated acetoxy-13α-triethylsiloxytaxa-4(20),11(12)-diene (**30**) (240 mg).

Tetrabutylammonium fluoride (1.0 mL, 1 M solution in THF, 1 mM) was dropped to the solution of compound **30** (240 mg, 0.5 mmol) in THF (5 mL) at room temperature and the reaction system was stirred with 1 h and quenched with water (10 mL). The organic phase was separated and the aqueous phase was extracted with EA (3 × 20 mL), the combined organic phase was washed with brine, dried with sodium sulfate, filtered and concentrated in vacuum. The resulting residue was purified on chromatography silica gel (PE/EA = 7/1) to obtain the desired deuterated substrate as colorless solid (150 mg, 67% yield for two steps).

### Chemical synthesis of $^{18}$O-labeled compound 23

To the solution of **29** (1.2 g, 2.87 mmol) and acetic acid-$^{18}$O$_2$ (732 mg, 4.6 mmol) in anhydrous dichloromethane (16 mL), the triethyl amine (2.0 mL, 14 mmol), 1,3-dicyclohexylcarbodiimide (1.18 g, 5.74 mmol), 4-dimethylaminopyridine (351 mg, 2.87 mmol) were added sequentially at room temperature. The resulted reaction mixture was stirred at room temperature with 2 days and then quenched by water (30 mL). The organic phase was separated and the aqueous phase was extracted with dichloromethane (3 × 40 mL). The combined organic phase was washed with brine, dried with anhydrous sodium sulfate, filtered and concentrated in vacuum. The resulted residue was purified on silica chromatography gel (PE/EA = 15/1) to obtain the inseparable mixture of **29** and $^{18}$O-labeled 5α-acetoxy-13α-triethylsiloxytaxa-4(20),11(12)-diene-13α-ol (**31**) (1.2 g).

The mixture of **29** and **31** was dissolved in anhydrous THF (10 mL) and Tetrabutylammonium fluoride (10.0 mL, 1 M solution in THF, 10.0 mmol) was dropped at room temperature. The reaction system was stirred at room temperature with 1 h and quenched with water (20 mL). The organic phase was separated and the aqueous phase was extracted with ethyl acetate (3 × 20 mL), the combined organic phase was washed with brine, dried with sodium sulfate, filtered and concentrated in vacuum. The resulted residue was purified on chromatography silica gel (PE/EA = 7/1) to obtain the desired $^{18}$O-labeled substrate **23** as colorless solid (422 mg, 42% yield for two steps).

### Chemical deacetylation of $^{18}$O-labeled 1β-dehydroxybaccatin VI (26a) to prepare $^{18}$O-labeled 4,7,9,10,13-pentadeacetyl-1β-dehydroxybaccatin VI (28a)

To the solution of $^{18}$O-labeled 1β-dehydroxybaccatin VI (**26a**) (2.4 mg, 3.4 μmol) in ethanol (200 μL), hydrazine hydrate (200 μL) was added. The resulted solution was stirred at room temperature over 32 h and then diluted with water (2 mL), neutralized with 1 M HCl aqueous solution[55]. The resulted aqueous solution was extracted with ethyl acetate (3 × 10 mL). The combined organic phase was washed with brine and dried with sodium sulfate, filtered and purified by prep-HPLC to obtain the desired $^{18}$O-labeled 4,7,9,10,13-pentadeacetyl-1β-dehydroxybaccatin VI (**28a**).

### Chemical introducing acetyl group to synthesize compounds 15, 20, and 25

The mixture of compound **14** (11.8 mg, 17.3 μmol) in acetic anhydride (0.5 mL) and pyridine (0.5 mL) was stirred at room temperature overnight and then concentrated in vacuum, the residue was purified by prep-HPLC to give the 2α-benzoyloxy-7β-acetoxytaxusin (**15**) (9.4 mg) in 85% yield. Similarly, 2α-benzoyloxy-5α-trideuterated acetyl-taxusin-7β-ol (**19**) (6.0 mg, 8.8 mmol) was stirred in acetic anhydride (0.5 mL) and pyridine (0.5 mL) to give 2α-benzoyloxy-7β-acetoxy-5α-trideuterated acetyl-taxusin (**20**) (4.8 mg) with 82% yield. $^{18}$O-labeled 2α-benzoyloxy-7β-acetoxytaxusin (**25**) (4.2 mg) was synthesized from $^{18}$O-labeled 2α-benzoyloxy-acetoxytaxusin7β-ol (**24**) (4.7 mg, 7.3 μmol) in 83% yield as the above procedure.

### Isolation and purification of standards

For the identification of products, the in vivo feeding systems were amplified correspondingly. Specifically, a single colony of yeast strain was incubated in 50 mL YPD medium in 250 mL shake flask and then grown at 30 °C, 250 rpm about 16 h. Then, 2 mL culture containing yeast strain was incubated in 200 mL medium in 1 L shake flask and yeast strain was grown at 30 °C, 250 rpm. In order to achieve the conversion efficiency in tube, low concentration substrate (1%, v/v, 2.5–50 μM in dimethyl sulfoxide) was added to the medium at 24 h. Product was extracted with EA at 60 h and purified with the silica gel column using different ratios of PE and EA as eluent. The purity of the product was judged by NMR analysis. If necessary, further purification was performed using Agilent 1260 Infinity II preparative liquid chromatograph system using a Welch Ulimate XB-C18 (5 μm, 21.2 mm × 20 mm) preparative column. Water (A) and acetonitrile (B) were used in the gradient elution system and products were detected at 203 nm. For the detail of isolation and purification of product, please see Supplementary Note 2.

### NMR analysis

All the $^1$H NMR, $^{13}$C NMR, 2D-COSY NMR, 2D-NOE NMR, 2D-HSQC NMR and 2D-HMBC NMR were recorded on Bruker AM-400, 500-MHz or Bruker AV Neo 500-, 600-MHz spectrometers in the CDCl$_3$, CD$_3$OD, C$_6$D$_6$ or D$_6$-DMSO and the data was analyzed and processed by using MestRenova (version 14.1.0). The chemical shift (δ) of $^1$H NMR was given in ppm relative to TMS (δ = 0.00 ppm), CHCl$_3$ (δ = 7.26 ppm), C$_6$H$_6$ (δ = 7.16 ppm) or DMSO (δ = 2.50 ppm). The chemical shift (δ) of $^{13}$C NMR was given in ppm relative to CDCl$_3$ (δ = 77.16 ppm), CD$_3$OD (δ = 49.00 ppm), C$_6$D$_6$ (δ = 128.06 ppm) or D$_6$-DMSO (δ = 39.52 ppm). All the $^2$H NMR were recorded on Bruker Bruker AM–400, 500 MHz spectrometers in CHCl$_3$, the chemical shift (δ) of $^2$H NMR was given in ppm relative to added standard CDCl$_3$ (δ = 7.26 ppm). The details of the NMR data including the methodologies for the determination the stereochemistry of key hydroxyl group and selectively acylation position were shown in Supplementary Note 2. All the NMR spectra and the key COSY, NOE and HSBC were supplied in Supplementary Figs. 13–115. NMR data of 1β-dehydroxybaccatin VI (**16**) is also available in Supplementary Table 6.

## Data availability

RNA sequencing data generated in this study have been deposited at NCBI under BioProject no. PRJNA1062083. CYP725A37, codon-optimized CYP725A37, CYP725A55 and AT5 have been deposited Genbank under accession numbers PP197199, PP197200, PP197201 and PP197202, respectively. Sequences of genes used in this study were obtained from Genbank under the following accession numbers: GGPPs (D28748 [https://www.ncbi.nlm.nih.gov/nuccore/D28748.1]), TS (U48796), T13OH (AY056019), T10OH (AF318211), T10OH2 (AY563635), T7OH (AY307951), T2OH (AY518383), T5AT (AF190130), TAX19 (AY628434), TCPR (AY571340), AAE4 (MN961507), and TBT (AF297618). Source data are provided with this paper.

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

## Acknowledgements

This work was financially supported by the National Key Research and Development Program of China (Grant No. 2018YFA0900700, Z.Z.), the Strategic Biological Resources Service Network Plan (Grant No. KFJ-BRP-009, Z.Z.), the Strategic Priority Research Program (Grant No. XDB27020206, Z.Z.) and International Partnership Program (153D31KYSB20170121, G.Z.) of the Chinese Academy of Sciences, and the National Natural Science Foundation of China (Grant No. 31921006, Z.Z.) as well as GsynBioT (Shanghai) Co., Ltd. We thank Prof. David Nelson from the committee of standardized cytochrome P450 Nomenclature for naming P450s.

## Author contributions

C.Y., Z.S., and P.W. performed the biological experiments; C.Y., Y.W., Z.S., L.X., and W.L. carried out the chemical experiments; C.Y. and X.Y. performed the bioinformatics analyses; X.Y., D.M., G.Z., and Z.Z. supervised and coordinated the experiments; C.Y., Y.W., D.M., G.Z., and Z.Z. wrote the manuscript. C.Y., Y.W., and Z.S. contributed equally to this work.

## Competing interests

These authors declare no competing interests.
