## [Peer Review File · Nature Communications]

Biosynthesis of the highly oxygenated tetracyclic core skeleton of TaxolEditorial Note: This manuscript has been previously reviewed at another journal that is not operating a transparent peer review scheme. This document only contains reviewer comments and rebuttal letters for versions considered at Nature Communications.

Reviewers' Comments:

Reviewer #1:

Remarks to the Author:

The authors have adequately addressed the concerns raised during the initial submission (provided below for convenience).

Minor note - After resubmission, Kampranis Lab demonstrated oxetane formation from taxadiene in a JACS publication via CYP725A4 (<https://doi.org/10.1021/jacs.3c10864>). The authors should comment on how their alternate route to the oxetane core compares to the mechanism established by Kampranis Lab, and reevaluate their experimental results in the context of this P450.

Previous comments from initial submission:

In this manuscript, Yang et al. report the elucidation of several missing enzymes involved in the biosynthesis of Taxol, enabling the partial pathway reconstitution to the newly proposed intermediate taxusin (12) as well as the oxetane-containing 1 β -dehydroxybaccatin VI (16). A key breakthrough was to leverage a hybrid synthetic chemistry-metabolic engineering approach, in which the chemically synthesized taxa-4(20),11(12)-diene-5 α -ol (2) was fed to modified yeast strains expressing downstream enzymes. This permitted stepwise investigation of the oxidation/acylation chemistry, including confirmation of previous pathway enzymes as well as identification of uncharacterized enzymes from transcriptomics data. Compounds are clearly labeled, and the structures look very consistent across figures. This systematic investigation also permitted the 11 step conversion of fed 2 to 1 β -dehydroxybaccatin VI, which will serve as a foundation for subsequent pathway elucidation studies.

Given the notoriety of this family of natural products, there should be great interest in this foundational pathway elucidation and strain engineering effort. Although the number of newly identified genes is limited to three P450s and an acyltransferase, the authors have provided complete spectroscopic data on all proposed intermediates. Moreover, a mechanistic investigation of the oxetane-forming P450 was performed, which sheds light on the biosynthetic route (direct oxidation > acyl rearrangement). One potential drawback which decreases the impact of this manuscript is that the work focuses on only a portion of the Taxol pathway. While the authors have investigated this section of the pathway quite comprehensively, this shortcoming could warrant publication in a less competitive journal. I would suggest the following major and minor revisions:

Major revisions:

>>>Synthetic biology details

The authors should expand on their Methods for yeast strain construction, as the constitutive expression of eleven genes (six of which are P450s) is no small task. Why were the selected promoters and terminators chosen? What were the markers that were used? Were markers recycled in such a heavily engineered strain? How many biological isolates were tested for each integration event?

The authors must also describe the exact amount of fed substrate (currently only a range is provided), and resulting titers for each product in grams per liter of culture for each strain. This information is critical for any synthetic biology study in order readers to assess the performance of the discovered and reconstituted enzymes in the chassis strain. The authors should describe the volumes of each culture required for isolation, and the quantities of each of the intermediates that were subjected to NMR analysis, as well as the culture methods (ie. What do the authors mean by "amplified correspondingly" on line 619?). This information is necessary for data reproducibility.

>>>Stereochemical determination

It is not clear to me how the hydroxy/acetoxy-group stereochemistry of the isolated intermediates was determined. The authors must clarify how the stereochemistry was spectroscopically confirmed. It would also help for the organic schema for all synthesized compounds to be presented – this would greatly facilitate manuscript review (ie. Understanding the stereospecific installation of the oxygen in the synthesized 2).

>>>Clarity of storytelling

The clarity and context of the story should be improved prior to publication. The authors should perform extensive copyediting to ensure the story is clear and concise. To name one specific example – the conversion of 7 to 8 by TAX19 seems quite low (Extended Data Fig 4e) relative to the conversion of 6 to 9 (Extended Data Fig 5b). Inclusion of this "alternative pathway" in Figure 2 may do more to confuse readers and could be better suited for the Supplementary Data. In addition to grammatical and readability improvements, the authors should consider modifying the figures to be more readable, especially for a general audience. As currently drawn, the structures are too small in Figures 1 and 4. Chromatogram traces should also be labeled and described in the captions. One last aesthetic note – the appearance of Figure 2 and 3 could be improved if the chromatograms and spectra were the same size.

Minor revisions

Line 136: The authors should explain what is meant by "unable or hard to produce" – ie provide the actual result from the citation in the main text to better justify their approach.

Line 228: The authors should provide these structures in a supplemental figure.

Line 256: The authors should provide the experimental details for synthesis of 20, including amounts of materials and yields.

Line 277: The authors should provide the experimental details for synthesis of 25, including amounts of materials and yields.

Line 305: The authors should comment on why compound 16 is the predominant product in YDBVI+2 cultures (with a small 17 shoulder – Fig. 4b) but appears in a 1:1 ratio with 17 in feeding studies using 15 and CYP725A-8 + TCPR (Fig. 3a).

Reviewer #2:

Remarks to the Author:

This manuscript has been revised to address my previously expressed concerns, other than presentation, which remains an issue (although not overwhelming). The authors have incorporated the suggestion to note and discuss the issues surrounding the actual identity of the first hydroxylated intermediate in Taxol biosynthesis. However, there has been an additional publication providing yet another alternative, formation of the oxetane ring by the relevant CYP725A4 (to taxologenic oxetane), which should also be noted and discussed, if nothing else for the suggestions that taxane biosynthesis is almost certainly a matrix rather than linear pathway (Zhao et al, 2023 JACS online). Otherwise, beyond the few minor changes listed below, no other issues are noted.

Minor corrections:

Please note CYP nomenclature is incorrect: "CYP725A-6" and "CYP725A-8" should be "CYP725A6" and "CYP725A8", respectively, throughout the presentation.

Line 268: "Compared" should be "Comparing"

Line 270: "was" should be deleted

Line 283: "labbeled" should be "labeled"

Line 295: "instead of the chemosynthesis" should be deleted

Line 324: "biosynthesis of 16" should be "biosynthesis of 16 from 2"

Line 327: "means of" should be deleted

Line 328: "made" should be "undertaken"

Line 331: add reference Guerra-Budd et al, 2012 Nat. Prod. Rep. 29:683

Line 336: alternative pathways from other early (initial hydroxylation) intermediates and possibility of metabolic matrix or grid for taxane biosynthesis should be noted and discussed here (with references)

Line 372: "labbeled" should be "labeled"

Line 424: taxologenic oxetane should be mentioned here (from Zhao et al, 2023)

Reviewer #3:

Remarks to the Author:

In this article, Zhou and co-workers identified three enzymes involved in Taxol biosynthesis through the analysis of the transcriptome data. These enzymes include a Taxoid 9 α -hydroxylase (CYP725A-6), an acetyl transferase catalyzing the formation of C7-OAc, and a P450 enzyme (CYP725A-8) catalyzing the formation of the oxetane ester in the taxoid tetracyclic core. The author conducted isotope labeling experiments to study the catalytic mechanism, demonstrating its formation through a novel cascade oxidation-acyl rearrangement mechanism in the oxetane ester of 1 β -dehydroxybaccatin VI. Using a combination of chemical synthesis and metabolic engineering, the author systematically studied both characterized and unidentified enzymes in vitro and in vivo. They determined the partial hydroxylation reaction sequence, elucidated the product formation, characterized the structures, and successfully achieved the biosynthesis of 1 β -dehydroxybaccatin VI from taxa-4(20),11(12)-diene-5 α -ol in engineered yeast. This work lays a foundation for the research on Taxol biosynthesis. Thus, I would like to suggest the publication in Nature Communications after minor revision.

Suggestions for modifications to the article:

Recently, relevant articles on paclitaxel biosynthesis have been published in Molecular Plant and JACS. It is advisable for the author to compare and discuss their findings with the existing literature in the discussion section.

The identified structures by the author seem reasonable, but for clearer presentation of structural elucidation, it is recommended that the author includes 2D NMR data (including COSY, HMBC, and

NOE correlations) for uncharacterized structures in the Supporting Information (SI). Additionally, some NMR data assignments appear to be incorrect; the chemical shift for 13 β -H at line 359 should be 4.39 instead of 4.10, and the chemical shift for 19-H at line 688 should be 1.36 instead of 1.57. The author should carefully review the data.

Mass spectrometry data in the main text lacks clarity. It is suggested to improve the resolution of the images.

In Extended Data Fig. 7, the labeling formats for hydrogen signals in figures b and f are inconsistent; it is recommended to standardize them. Additionally, explanations for the remaining differential peaks in figures c and d should be provided.

When determining the stereochemistry of compound 13, the NOE correlation between 2 β -H (δ 4.26) and 1 β -H (δ 2.14) does not explain the configuration of C10-OH. More data should be presented to clarify its configuration. Moreover, in the characterization of the stereochemistry of compound 17, simultaneous observation of NOE correlations between 3 α -H (δ 3.08) / 20 α -H (δ 3.58) and 5 β -H (δ 4.20) / 20 β -H (δ 2.26) seems contradictory. The author should further confirm the structure.

Response to reviewers

We would like to sincerely thank reviewers' constructive comments and helpful suggestions to improve our manuscript. We have revised the manuscript according to the Reviewers' suggestions, and have addressed all of the concerns in the point to point response letter

Reviewer #1 (Remarks to the Author):

Q1:The authors have adequately addressed the concerns raised during the initial submission.

Minor note - After resubmission, Kampranis Lab demonstrated oxetane formation from taxadiene in a JACS publication *via* CYP725A4 (<https://doi.org/10.1021/jacs.3c10864>). The authors should comment on how their alternate route to the oxetane core compares to the mechanism established by Kampranis Lab, and reevaluate their experimental results in the context of this P450.

>Response: Thanks for your positive comments and constructive suggestions to improve our manuscript. According to your suggestions, we added several sentences to compare similar or alternative catalysis processes for the formation of oxetane from three independent studies in discussion as following (Lines 436-447):

Recently, T5OH was proposed to catalyze oxetane ring formation *via* the rearrangement of unstable epoxide intermediate from endotaxadiene and exotaxadiene based on the detection of a taxologenic oxetane among the newly found four products catalyzed by T5OH⁴⁰. In another study⁴¹, taxologenic oxetane formation was also proposed to be catalyzed by T5OH together with a 2-oxoglutarate/Fe(II)-dependent dioxygenase. Although these proposed mechanisms, different from that of the cascade oxidation-concerted acyl rearrangement catalyzed by CYP725A55 shown in this study and Jiang *et al.*⁴², need further experimental confirmations, the presence of both multi-products of T5OH and different mechanisms potential for oxetane ring formation implicated that the biosynthetic pathway of Taxol is more likely a complex, matrix rather than a linear pathway, and future comprehensive verifications are needed.

Reviewer #2 (Remarks to the Author):

This manuscript has been revised to address my previously expressed concerns, other than presentation, which remains an issue (although not overwhelming). The authors have incorporated the suggestion to note and discuss the issues surrounding the actual identity of the first hydroxylated intermediate in Taxol biosynthesis.

Q2: However, there has been an additional publication providing yet another alternative, formation of the oxetane ring by the relevant CYP725A4 (to taxologenic oxetane), which should also be noted and discussed, if nothing else for the suggestions that taxane biosynthesis is almost certainly a matrix rather than linear pathway (Zhao et al, 2023 JACS online). Otherwise, beyond the few minor changes listed below, no other issues are noted. **Q12:** Line 336: alternative pathways from other early (initial hydroxylation) intermediates and possibility of metabolic matrix or grid for taxane biosynthesis should be noted and discussed here (with references). **Q14:** Line 424: taxologenic oxetane should be mentioned here (from Zhao et al, 2023)

>Response: Thanks for your positive comments and constructive suggestions. All of these three suggestions were based on the recent three independent studies (MP, JACS and Science). So we have rewrote discussion section by discussing similar or alternative catalysis processes for the oxetane ring formation from recent three independent studies (MP, JACS and Science). Several sentences to describe the matrix biosynthetic pathways of Taxol and compare similar or alternative catalysis processes for the formation of oxetane from three independent studies were added in discussion section as following (Lines 436-447):

Recently, T5OH was proposed to catalyze oxetane ring formation *via* the rearrangement of unstable epoxide intermediate from endotaxadiene and exotaxadiene based on the detection of a taxologenic oxetane among the newly found four products catalyzed by T5OH⁴⁰. In another study⁴¹, taxologenic oxetane formation was also proposed to be catalyzed by T5OH together with a 2-oxoglutarate/Fe(II)-dependent dioxygenase. Although these proposed mechanisms, different from that of the cascade oxidation-concerted acyl rearrangement catalyzed by CYP725A55 shown in this study and Jiang *et al.*⁴², need further experimental confirmations, the presence of both multi-

products of T5OH and different mechanisms potential for oxetane ring formation implicated that the biosynthetic pathway of Taxol is more likely a complex, matrix rather than a linear pathway, and future comprehensive verifications are needed.

Minor corrections:

Q3: Please note CYP nomenclature is incorrect: “CYP725A-6” and “CYP725A-8” should be “CYP725A6” and “CYP725A8”, respectively, throughout the presentation.

>Response: Thanks for your reminding. CYP725A-6 and CYP725A-8 were named by us and informal. We have asked David Nelson (the committee of standardized cytochrome P450 Nomenclature) to give their formal names. David Nelson named CYP725A-6 and CYP725A-8 as CYP725A37 and CYP725A55, respectively. We have changed as these names throughout full text and related files.

Q4: Line 268: “Compared” should be “Comparing”

>Response: Thank you, we have revised it.

Q5: Line 270: “was” should be deleted

>Response: Thank you, we have deleted it.

Q6: Line 283: “labbeled” should be “labeled”

>Response: Thank you, we have revised it.

Q7: Line 295: “instead of the chemosynthesis” should be deleted

>Response: Thank you, we have deleted it.

Q8: Line 324: “biosynthesis of 16” should be “biosynthesis of 16 from 2”

>Response: Thank you, we have revised it.

Q9: Line 327: “means of” should be deleted

>Response: Thank you, we have deleted it.

Q10: Line 328: “made” should be “undertaken”

>Response: Thank you, we have revised it.

Q11: Line 331: add reference Guerra-Budd et al, 2012 Nat. Prod. Rep. 29:683

>Response: Thank you, we have added this reference.

Q13:Line 372: “labbeled” should be “labeled”

>Response: Thank you, we have revised it.

Reviewer #3 (Remarks to the Author):

In this article, Zhou and co-workers identified three enzymes involved in Taxol biosynthesis through the analysis of the transcriptome data. These enzymes include a Taxoid 9 α -hydroxylase (CYP725A-6), an acetyl transferase catalyzing the formation of C7-OAc, and a P450 enzyme (CYP725A-8) catalyzing the formation of the oxetane ester in the taxoid tetracyclic core. The author conducted isotope labeling experiments to study the catalytic mechanism, demonstrating its formation through a novel cascade oxidation-acyl rearrangement mechanism in the oxetane ester of 1 β -dehydroxybaccatin VI. Using a combination of chemical synthesis and metabolic engineering, the author systematically studied both characterized and unidentified enzymes in vitro and in vivo. They determined the partial hydroxylation reaction sequence, elucidated the product formation, characterized the structures, and successfully achieved the biosynthesis of 1 β -dehydroxybaccatin VI from taxa-4(20),11(12)-diene-5 α -ol in engineered yeast. This work lays a foundation for the research on Taxol biosynthesis. Thus, I would like to suggest the publication in Nature Communications after minor revision.

Suggestions for modifications to the article:

Q14: Recently, relevant articles on paclitaxel biosynthesis have been published in Molecular Plant and JACS. It is advisable for the author to compare and discuss their

findings with the existing literature in the discussion section.

>Response: Thanks for your positive comments and constructive suggestions to improve our manuscript. We added several sentences to describe the matrix biosynthetic pathways of Taxol and these different routes to the formation of oxetane in discussion section as following (Lines 335-345, 390-393, 436-447 and 448-460):

Lines 335-345: Followed by step-wise re-characterizing key enzymes or identifying novel enzymes *via* integrated assays of *in vivo* substrate feeding of cells and *in vitro* substrate incubation with crude protein lysates based on their corresponding gene-expressed *S. cerevisiae* chassis cells, pathways for biosynthesis of the highly oxygenated tetracyclic core skeleton of Taxol, 1 β -dehydroxybaccatin VI (**16**) from **2** was elucidated (Fig 4). Along with the reviewing process of this manuscript, similar or alternative catalysis processes for the oxetane ring formation were published by three independent studies⁴⁰⁻⁴², of which, the study of Jiang *et al.*⁴² not only supported our findings of enzyme characterization but also proved that this biosynthesis pathway network elucidated *via S. cerevisiae* chassis cells is largely similar to that *via* the tobacco chassis.

Lines 390-393: It is interesting to notice that Jiang *et al.* also proposed an oxetane ring formation mechanism of oxidation-acyl rearrangement without using epoxide as an intermediate based on density functional theory (DFT) calculations⁴², supporting our cascade oxidation-concerted acyl rearrangement mechanism.

Lines 436-447: Recently, T5OH was proposed to catalyze oxetane ring formation *via* the rearrangement of unstable epoxide intermediate from endotaxadiene and exotaxadiene based on the detection of a taxologenic oxetane among the newly found four products catalyzed by T5OH⁴⁰. In another study⁴¹, taxologenic oxetane formation was also proposed to be catalyzed by T5OH together with a 2-oxoglutarate/Fe(II)-dependent dioxygenase. Although these proposed mechanisms, different from that of the cascade oxidation-concerted acyl rearrangement catalyzed by CYP725A55 shown in this study and Jiang *et al.*⁴², need further experimental confirmations, the presence of both multi-products of T5OH and different mechanisms potential for oxetane ring formation implicated that the biosynthetic pathway of Taxol is more likely a complex,

matrix rather than a linear pathway, and future comprehensive verifications are needed.

Lines 448-460: In summary, our systematic studies clearly elucidated the complete biosynthetic route to the highly-oxygenated 1β -dehydroxybaccatin VI with the Taxol tetracyclic core skeleton from the mono-oxygenated taxoid taxa-4(20),11(12)-diene- 5α -ol, laying down a solid foundation for the breakthrough of complete elucidation of the biosynthetic pathway of Taxol. Of course, further efforts are required to elucidate the complete biosynthetic pathway of baccatin III from 1β -dehydroxybaccatin VI and other parallel biosynthetic pathway of Taxol. It is also interesting to further confirm whether Taxoid 1β -hydroxylase and 9α -dioxygenase are necessary⁴¹ or unwanted⁴² to biosynthesize baccatin III in *N. benthamiana* chassis and *Taxus* plants in the future. While an important experience learnt from the recent emergence of breakthroughs⁴⁰⁻⁴², including this study, cross-disciplinary investigations and collaborations, particularly between the chemists and biochemists with the phytochemists and plant molecular physiologists are essential in order to completely fulfill this long-awaited endeavor.

Q15: The identified structures by the author seem reasonable, but for clearer presentation of structural elucidation, it is recommended that the author includes 2D NMR data (including COSY, HMBC, and NOE correlations) for uncharacterized structures in the Supporting Information (SI). Additionally, some NMR data assignments appear to be incorrect; the chemical shift for 13β -H at line 359 should be 4.39 instead of 4.10, and the chemical shift for 19-H at line 688 should be 1.36 instead of 1.57. The author should carefully review the data.

>Response: Thanks for your careful reviewing and constructive advices. The key 2D NMR correlations including COSY, NOE and HMBC for the newly characterized compounds were shown in the NMR spectra section of SI. All the data were rechecked and the mistyped chemical shifts were corrected in the revised supporting information. The chemical shift for 13β -H in compound **5** was corrected to 4.39 (Page S27) and the chemical shift for 19-H in compound **22** was also corrected to 1.36 (Page S37).

Q16: Mass spectrometry data in the main text lacks clarity. It is suggested to improve the resolution of the images.

>Response: Thanks. We have improved the resolution of mass spectrometry data by adjusting the format of text and lines. We will finally offer separated high resolution original figures (Tiff format) to this journal instead of this incorporated PDF type.

Q17: In Extended Data Fig. 7, the labeling formats for hydrogen signals in figures b and f are inconsistent; it is recommended to standardize them. Additionally, explanations for the remaining differential peaks in figures c and d should be provided.

>Response: Thanks for your reminding. We have standardized the labeling formats for hydrogen signals Extended Data Fig. 7. The limited amount of acetyl-CoA in *in vitro* reaction (Extended Data Fig. 7d) may mainly contribute to the differential peaks in *in vitro* and *in vivo* reaction (Extended Data Fig. 7c and d). These peaks were proposed as the monoacetylated **11** at C9-OH or C10-OH. We have revised it as following (Lines 216-217):

Thus, when **11** was employed as the substrate for both *in vivo* and *in vitro* reactions using the *S. cerevisiae* strain expressing T5AT, HPLC analysis demonstrated that **12** was formed, accompanied with proposed monoacetylated **11** at C9-OH or C10-OH (Extended Data Fig. 7c-g), confirming T5AT could add acetyl groups towards the C9-OH and C10-OH of **11** to form **12**.

Q18: When determining the stereochemistry of compound **13**, the NOE correlation between 2 β -H (δ 4.26) and 1 β -H (δ 2.14) does not explain the configuration of C10-OH. More data should be presented to clarify its configuration. Moreover, in the characterization of the stereochemistry of compound **17**, simultaneous observation of NOE correlations between 3 α -H (δ 3.08) / 20 α -H (δ 3.58) and 5 β -H (δ 4.20) / 20 β -H (δ 2.26) seems contradictory. The author should further confirm the structure.

>Response: Thanks for your advice. The NOE correlation of compound **13** was added in the NMR section of SI (Page S76) and the corresponding description were added in the NMR data section of SI (Page S34).

The stereochemistry of newly introduced C7-hydroxy were confirmed by 2D-NOE ($7\alpha\text{-H}$ (δ 4.23)/ $3\alpha\text{-H}$ (δ 2.96)), there is a strong NOE correlation between $7\alpha\text{-H}$ (δ 4.23)/ $10\alpha\text{-H}$ (δ 6.18) implied the stereochemistry of C10-hydroxy was resistant and still in *beta* position. The NOE correlation of $2\beta\text{-H}$ (δ 4.26)/ 19-H (δ 0.97) indicated the hydroxyl group in C2 was in *alpha* position.

The key NOE correlation of compound **17** was shown in page S82 in SI and the corresponding description in page S37 were changed as follows:

$5\beta\text{-H}$ (δ 4.20) and 20b-H (δ 2.26) has middle NOE correlation (The distance of the two hydrogen is calculated as 2.41 Å).The distance of 20a-H (δ 3.58) and $3\alpha\text{-H}$ (δ 3.08) is about 3.19 Å, so these two hydrogens have a weak NOE correlation. The 20a-H (δ 3.58) also shows strong NOE correlations with $14\alpha\text{-H}$ (δ 1.66) and Ph-H (δ 7.94). All those NOE correlations indicated the newly introduced C4(20) epoxide was in *beta* position.

REVIEWERS' COMMENTS

Reviewer #1 (Remarks to the Author):

All concerns have been addressed.

Reviewer #2 (Remarks to the Author):

This has been revised and adequately addresses my previously expressed concerns

Reviewer #3 (Remarks to the Author):

All concerns have been addressed. Now this manuscript is acceptable for publication.

REVIEWERS' COMMENTS

Reviewer #1 (Remarks to the Author): All concerns have been addressed.

Reviewer #2 (Remarks to the Author): This has been revised and adequately addresses my previously expressed concerns

Reviewer #3 (Remarks to the Author): All concerns have been addressed. Now this manuscript is acceptable for publication.

Response: We really appreciate all the reviewers for their efforts in improving our manuscript.